# Lower airway microbiota and decreasing lung function in young Brazilian cystic fibrosis patients with pulmonary *Staphylococcus* and *Pseudomonas* infection

Paulo Kussek[1], Dany Mesa[2,3]*, Thaís Muniz Vasconcelos[3], Luiza Souza Rodrigues[3], Damaris Krul[3], Humberto Ibanez[2], Helisson Faoro[4], Jussara Kasuko Palmeiro[5], Libera Maria Dalla Costa[3]*

1 Hospital Pequeno Príncipe, Curitiba, Paraná, Brazil, 2 Big Data Center, Instituto de Pesquisa Pelé Pequeno Príncipe, Curitiba, Paraná, Brazil, 3 Instituto de Pesquisa Pelé Pequeno Príncipe, Curitiba, Paraná, Brazil, 4 Instituto Carlos Chagas, Curitiba, Paraná, Brazil, 5 Departamento de Análises Clínicas, Centro de Ciências da Saúde, Universidade Federal de Santa Catarina, Florianópolis, Santa Catarina, Brazil

* lmdallacosta@gmail.com (LMDC); dmesaf7@gmail.com (DM)

**Data Availability Statement:** All relevant data are within the paper and its Supporting Information files.

## Abstract

Cystic fibrosis (CF) is a genetic disease caused by mutations in the cystic fibrosis transmembrane conductance regulator gene that leads to respiratory complications and mortality. Studies have shown shifts in the respiratory microbiota during disease progression in individuals with CF. In addition, CF patients experience short cycles of acute intermittent aggravations of symptoms called pulmonary exacerbations, which may be characterized by a decrease in lung function and weight loss. The resident microbiota become imbalanced, promoting biofilm formation, and reducing the effectiveness of therapy. The aim of this study was to monitor patients aged 8–23 years with CF to evaluate their lower respiratory microbiota using 16S rRNA sequencing. The most predominant pathogens observed in microbiota, *Staphylococcus* (Staph) and *Pseudomonas* (Pseud) were correlated with clinical variables, and the *in vitro* capacity of biofilm formation for these pathogens was tested. A group of 34 patients was followed up for 84 days, and 306 sputum samples were collected and sequenced. Clustering of microbiota by predominant pathogen showed that children with more Staph had reduced forced expiratory volume in one second (FEV$_1$) and forced vital capacity (FVC) compared to children with Pseud. Furthermore, the patients' clinical condition was consistent with the results of pulmonary function. More patients with pulmonary exacerbation were observed in the Staph group than in the Pseud group, as confirmed by lower body mass index and pulmonary function. Additionally, prediction of bacterial functional profiles identified genes encoding key enzymes involved in virulence pathways in the Pseud group. Importantly, this study is the first Brazilian study to assess the lower respiratory microbiota in a significant group of young CF patients. In this sense, the data collected for this study on the microbiota of children in Brazil with CF provide a valuable contribution to the knowledge in the field.

**Funding:** This study was fully funded by our institution (Instituto de Pesquisa Pelé Pequeno Príncipe). The funders had no role in study design, data collection and analysis, decision to publish, or preparation of the manuscript.

**Competing interests:** The authors have declared that no competing interests exist.

## Introduction

Cystic fibrosis (CF) is the most common life-shortening rare disease with an estimated incidence of 1 in every 6000 live births in Euro-Brazilians and 1/14000 in Afro-Brazilians [1]. The disease is caused by mutations in the CF transmembrane conductance regulator gene (CFTR), and the homozygous F508del is present in approximately 48% of all CF alleles [2]. Complications of CF disease begin in early life and over time, a combination of impaired mucociliary clearance, innate immune responses, inflammatory pulmonary process, chronic infection leads to bronchiectasis and respiratory failure [3].

The microbiota of the respiratory tract is recognized as an essential factor in the homeostasis of the respiratory system [4]. The respiratory microbiota is linked to progressive CF lung disease depending on many factors such as the time of diagnosis, patient age, chronic use of antibiotics, and mutation type of the CFTR gene [5]. Thus, the establishment of a community composed mainly of typical CF pathogens with other agents such as anaerobic bacteria, fungi, and viruses may cause dysbiosis of the respiratory system [5]. In addition, the pathophysiology of CF affects the respiratory microbiota, with the formation of biofilm and mucus plugging, making the pulmonary distal airways inaccessible by treatment agents [6].

A wide variety of bacterial species can be identified from patients with CF; the most frequently observed include *Staphylococcus aureus*, *Pseudomonas aeruginosa*, *Haemophilus influenzae*, and *Burkholderia cepacia* complex [7, 8]. Other opportunistic bacterial species that are less frequently detected in CF patients include *Stenotrophomonas maltophilia*, *Achromobacter xylosoxidans*, *Ralstonia* spp., *Pandoraea* spp., *Cupriavidus* spp., and non-tuberculosis mycobacteria [9, 10]. In addition, CF patients experience short cycles of acute intermittent aggravations of symptoms called pulmonary exacerbations, characterized by a decrease in lung function and weight loss, generally caused by opportunistic pathogens which can promote biofilm formation and reduce the effectiveness of therapy [11].

Nevertheless, the composition of respiratory microbiota varies noticeably among individuals; some patients show marked changes in the bacterial community with alternating infectious agents, and others show community resilience [3]. We analyzed the microbiota of 306 sputum samples of patients with CF and evaluated correlations with clinical variables (mutation type and patient's clinical status). In addition, we grouped CF patients in two groups by the dominant respiratory microbiota pathogens, Staph and Pseud, and analyzed these groups by their clinical variables using Pearson's correlation analysis and non-metric multidimensional scaling. Furthermore, bacterial functional profiles were predicted for both pathogen groups.

## Material and methods

### Study setting

This study was performed at the Pequeno Príncipe Hospital, the largest pediatric hospital in Brazil. Currently, 390 pediatric beds are available in 32 pediatric specializations. The CF unit includes 80 pediatric patients who are followed until they are transferred to an adult unit.

### Study population and clinical data

In this study, a group of 34 CF patients aged between 8 and 23 years was followed for 84 d (**Fig 1**). Patients were diagnosed by a sweat test and CFTR gene screening. After each regularly scheduled clinic visit or hospital admission, clinical data including the use of broad- and narrow-spectrum antibiotics, body mass index (BMI), and lung function parameters such as forced expiratory volume in one second ($FEV_1$) and forced vital capacity (FVC) were collected. Lung function was assessed using a JAEGER MasterScope® Spirometer (VIASYS Healthcare

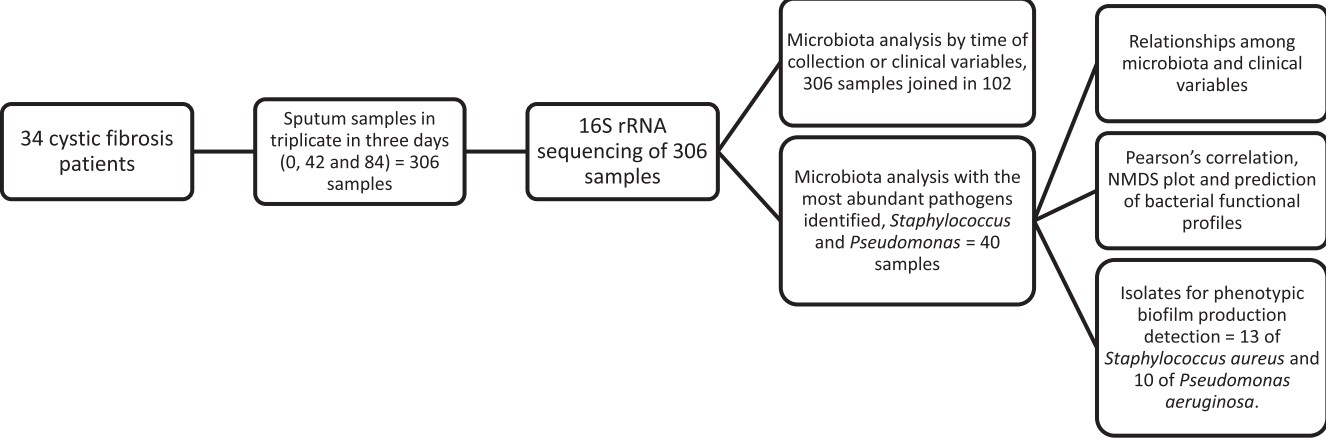

**Fig 1. Flowchart of samples analyzed in the study.**

GmbH, Hoechberg, Germany) following the standardization of pulmonary function test (PFT) by ATS/ERS Task Force [12]. Patients with $O_2$ saturation ≥88% in room air and capacity to perform PFT were included. The clinical conditions were categorized as baseline, exacerbation, treatment, and recovery [10]. Baseline condition: no acute respiratory symptoms and no systemic antibiotic use for >30 d; patients may or may not be on maintenance antibiotics such as azithromycin or inhaled antibiotics. Exacerbation condition: the initiation of acute respiratory symptoms and use of antibiotics (oral or intravenous). Treatment condition: the use of intravenous or oral antibiotics for more than 7 d for pulmonary exacerbation treatment. Recovery: no systemic antibiotic use for >7 d; patients may or may not be on maintenance antibiotics and may or may not be back to the baseline clinical condition.

## Ethics statement

The Institutional Review Board (IRB) of the participating center (IRB #2.405.167) approved this study, and informed consent was obtained from the parents or guardians of all participants. Research was conducted in a manner to ensure the confidentiality for each patient.

## Sample collection, processing, bacterial culture, and identification

A triplicate of sputum samples of the patients were collected on days 0, 42, and 84 (a total of 306 samples). Inhalation of hypertonic sterile saline solution (7%) by nebulization was used for sputum collection, followed by coughing and expectoration of airway secretions. Sputum characteristics ranged from salivary to purulent. The collected sputum samples were transported to the microbiology laboratory for processing within 2 h. Sputum samples were transferred to 15 mL graduated Falcon tubes, free of DNase and RNase, and sterile phosphate-buffered saline was added to bring the total volume to 8 mL. After homogenization, 2 mL of purulent sputum was transferred to new tubes and treated with β-mercaptoethanol and DNase I (Sigma-Aldrich, St. Louis, United States) to remove proteins and other soluble DNA, such as mitochondrial DNA [13]. The obtained pellet after treatment of all purulent samples and 1 mL aliquots of saliva samples were stored in a freezer at -80°C until DNA extraction. The remaining volume in the initial Falcon tubes was sent to the microbiology laboratory for bacterial culture identification [14]. *S. aureus* and *P. aeruginosa* isolated from sputum samples and identified by matrix-assisted laser desorption ionization mass spectrometry (MALDI-TOF MS) using a MicroflexTM LT instrument (Bruker Daltonics, Billerica, MA, USA) were stored at -80°C in

brain heart infusion broth (HIMEDIA, Mumbai, Maharashtra, India) with 20% (v/v) glycerol for further analysis [15].

## Phenotypic biofilm production detection

Qualitative biofilm production was performed using the tube method previously described [16]. A loop of microorganisms collected from tryptone soya agar (OXOID, Basingstoke, Hampshire, England) was inoculated into a polystyrene tube (15 mL Falcon tube) containing 10 mL of tryptone soy broth (OXOID, Basingstoke, Hampshire, England) supplemented with glucose (final concentration of 8%). Tubes were incubated at 35 ± 2˚C for 24 h, and the broth was gently aspirated. The tubes were washed thoroughly with phosphate-buffered saline (pH 7.2) and dried. Cells in the dried tubes were stained with 0.1% crystal violet for 7 min, and excess dye was removed by washing the cells with distilled water [16]. After drying, the tubes were visually evaluated for biofilm formation (presence or absence). Biofilm formation was considered positive when a visible film coated the wall and bottom of the tube. The experiments were performed in duplicate, and biofilm production was evaluated independently by two different observers. The sterile tryptone soy broth supplemented with glucose and the non-biofilm producer *Candida albicans* were used as a negative control and the biofilm producer *Candida tropicalis* was used as a positive control in the assay.

## DNA extraction and 16S rRNA amplicon sequencing

All samples (frozen pellet and saliva) were kept on ice until they were completely thawed when subjected to genomic DNA extraction. A volume of 750 μL of lysis buffer was added to the pellet of each purulent sample and homogenized until the pellet was dissolved. The same volume of lysis buffer (750 μL) was added to 250 μL of each saliva sample. The total volume of each mixture was transferred to a ZR BashingBead® Lysis tube (Zymo Research, CA, USA) for DNA extraction. DNA extraction was performed using the ZymoBIOMICS® DNA Miniprep Kit (Zymo Research, CA, USA), according to the manufacturer's recommendations. The purity and quality of the DNA were verified using a NanoVue Plus spectrophotometer (GE Healthcare, Life Sciences, Marlborough, MA, USA). Subsequently, DNA was stored at -80˚C.

Polymerase chain reaction (PCR) and universal primers (F515/R806) were used to amplify the V4 region of the 16S rRNA gene [17]. PCR consisted of 2.5 μL bovine serum albumin (3 mg/mL), 2.5 μL high-fidelity buffer (10x), 0.63 μL $MgCl_2$ (50 mM), 0.50 μL of dNTPs (10 mM), 0.625 μL primer mix (10 mM), 0.125 μL high-fidelity Taq polymerase (5 U/μL), 10 ng of template DNA, and 16.12 μL ultrapure water added to obtain a volume of 25 μL. The reaction conditions were as follows: 5 min at 95˚C, 25 cycles of 40 s at 95˚C, 2 min at 64˚C, 1 min at 72˚C, and 10 min at 72˚C. The amplicons were quantified with Qubit using an HS dsDNA kit (Invitrogen, Carlsbad, CA, USA), diluted to 500 pM, and pooled. Next, 16 pM of pooled DNA was sequenced using the MiSeq reagent 600V3 (Illumina, San Diego, CA, USA). Sequencing was performed using a MiSeq® sequencer (Illumina) to obtain paired reads of 250 bp [18]. A negative control for sequencing was used to check contamination.

## Sequencing data and statistical analysis

Sequencing data were analyzed using the QIIME2 Core 2021.8 pipeline [19]. Triplicate paired reads of the same collection of patients were joined in a single file (total of 102 samples). Next, the merged samples were filtered by quality, chimeras were removed and clustered into amplicon sequence variants using the DADA2 algorithm [20] in the QIIME2 program. Subsequently, taxonomic assignment was performed using the SILVA database, release 138 [21].

The reads output was normalized to 51,000 per sample, allowing a comparison of alpha and beta diversity between the groups.

Analysis of microbiota by the time of collection and clinical and genetic variables was compared using Welch's t-test ($P < 0.05$) and Bonferroni correction in STAMP software. Next, the abundance of bacterial taxa was compared for different pathogen groups using Welch's t-test ($P < 0.05$) and Bonferroni correction in STAMP software [22]. Clinical variables were analyzed using the Kruskal–Wallis and Mann-Whitney test ($P < 0.05$). Pearson's correlation coefficients and non-metric multidimensional scaling (NMDS) plots were calculated using the psych and vegan packages included in R software [23], and functional profiles of Staph and Pseud were obtained using the Tax4Fun program [24]. Only statistically significant results were reported ($P < 0.05$).

## Data accessibility

The dataset was submitted to the National Center for Biotechnology Information (NCBI) database under the BioSample accession code SAMN19760689.

## Results

### Study population and clinical data

Of the 34 study participants, sputum samples were collected on days 0, 42, and 84 (a total of 102 samples), 17 were male, with a mean age of 15.3 years (range of 8–23 years), and 31 had an early diagnosis before age 2. All the patients underwent PFT (spirometry) with a wide range of the impaired pulmonary function with FVC ranging from 31% to 152% (mean = 84.3%) and $FEV_1$ from 26% to 157% (mean = 75.9%). As for the nutritional status (BMI and Z score), the most repeated value in three collections was sought: 10 patients were considered eutrophic [25], 16 patients had grade I malnutrition, 8 patients had grade II malnutrition, and 2 patients had grade III malnutrition [26]; however, 12 individuals had score differences between collections for a higher or lower standard deviation according to their clinical condition at the time. Of the 34 patients, 26 had a negative Z-score (mean = -0.59), and 32 patients had pancreatic insufficiency. *S. aureus* was the most frequent microorganism (47%) identified by the culture-based methods, followed by *P. aeruginosa* (16%); both microorganisms were present in association from different culture media (25%), and in negative culture (14%). The metadata content of the clinical variables of the patients enrolled in this study is shown in supporting information (**S1 Table**).

### Phenotypic biofilm production detection

Sputum samples that showed Staph and Pseud with relative abundance >50% in the 16S rRNA sequencing were stored in -80˚C for further biofilm production (total 40 samples, 20 samples of each pathogen). Therefore, a total of 13 *S. aureus* and 10 *P. aeruginosa* strains were evaluated for their biofilm production capacity. All of these were *in vitro* biofilm producers. Only the dichotomous analysis, presence or absence, was used, considering that by visual analysis the interpretation of intensity (+/++ and +++) can be subjective, especially without specific controls for each of these categories.

### Taxonomic classification of sputum microbiota

This section presents the results of the analysis carried out at the genera and species levels. The analysis of the bacterial community at different collection times (days 0, 42, and 84) revealed 226 taxa, distributed among 178 genera and 48 species. The most abundant genera in the

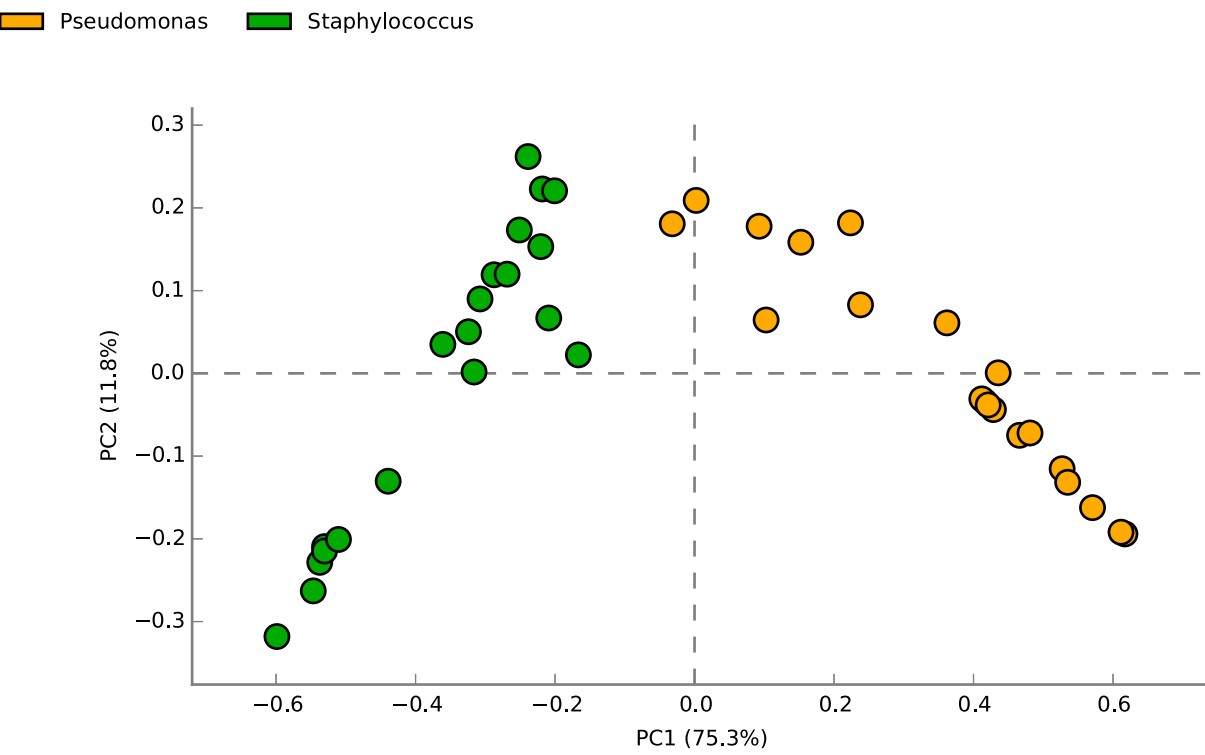

**Fig 2. Beta diversity of bacterial community, represented by PCA plot.** Dots with the same color mean samples of the same group, n = 20 per group.

community were *Veillonella*, *Prevotella*, *Haemophilus*, *Pseudomonas*, *Staphylococcus*, *Strepto-coccus*, *Serratia*, *Neisseria*, and *Porphyromonas* (**S1 Fig**). A total of 155 taxa represented the core microbiota of the community on days 0, 42, and 84; these taxa were identified at all sampling periods and in all samples (**S2 Fig**).

Analyses of microbiota considering the time of collection, clinical variables or mutation did not identify any specific pattern in the lower respiratory microbiota of CF patients associated with the different clinical conditions. Thus, considering this result, the samples were clustered into two main groups Staph and Pseud, based on culture results and the most abundant pathogens identified in 16S rRNA sequencing (abundance >50%). Few samples with great abundance for other genera such as *Veillonella* and *Haemophilus*, among others were observed. Owing to the low number of samples, other microorganisms were not analyzed. The clinical variables of the patients were correlated with these two pathogen groups. Beta diversity analysis showed an adequate clustering of groups by pathogen type, which was reflected in the results of the principal component analysis (PCA), which highlighted that each group had a dominant organism, Staph or Pseud (**Fig 2**).

## Correlations among clinical variables and microbiota

We analyzed the clinical data by grouping patients according to the type of predominant pathogen in the sputum microbiota (the selection criterion was patients with pathogen abundance >50% in microbiota). We identified two dominant groups, Staph and Pseud, with a genetic mutation frequency of F508del in 67% of alleles, followed by G542x in 20%, 1078delT in 6%, and others (R334W and 2184delA) in 7% of alleles. The analyses revealed the following results:

The BMI was obtained by comparing patients of the same age; the BMI values were 18.09 and 19.41 for the Staph and Pseud groups, respectively. In other words, the patients in the Staph group were underweight (BMI $<18.5$ kg/m$^2$ is considered underweight). On the other hand, the patients in the Pseud group had normal weight (BMI = 18.5–25.0). All patients in the Pseud group (except one patient) received antimicrobial treatment against *Pseudomonas* spp., in other words, chronically inhaled antibiotic to reduce bacterial growth and the frequency of exacerbations. In addition, we chose BMI values because this parameter had positive values. Besides, BMI Z-scores had negative values, which made it difficult to use Pearson's correlation analysis or any other statistical analysis.

The clinical condition of the patients was as follows: in the Staph group, nine patients were classified in the baseline clinical condition, five in the exacerbated, six in the treatment, and one in the recovery group. In the Pseud group, 11 patients were classified in the baseline clinical condition, one in exacerbated, and five in the treatment group. Due to chronic lung infection, most CF patients had reduced lung function with significant differences ($P <0.05$) between the two groups; in general, the Staph group had reduced FEV$_1$ and FVC compared to the Pseud group (**Fig 3**).

## Pearson's correlation analysis between clinical variables and microbiota

Pearson's correlation coefficient showed significant relationship ($P <0.05$) between microbiota and clinical variables. In the Staph group, there were significant negative correlations between the variables (FVC and FEV) and prevalence of Staph and *S. aureus* (**Fig 4**). In contrast, positive correlations were observed between clinical variables (BMI and age; FVC and FEV). Similarly, positive correlations were observed between the pathogens Staph and *S. aureus*. These results highlight the veracity of the correlation matrix. On the other hand, in the Pseud group, there was no significant positive or negative relationship between microbiota and clinical variables.

## Non-metric multidimensional scaling (NMDS)

To represent the behavior of the variables in a multivariate system, we used an NMDS plot. This analysis reveals the pairwise dissimilarity between objects in a two-dimensional space, in this case, microbiota and clinical variables (plotted as vectors). NMDS results confirmed the clustering of the pathogens in two groups with different taxonomic compositions. The first

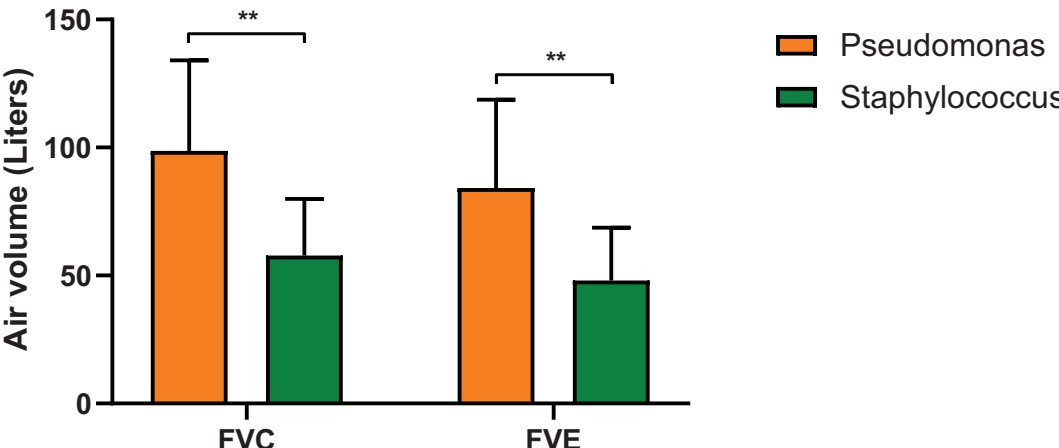

**Fig 3. Comparison between pathogens showing decreased forced vital capacity and forced expiratory volume in patients in the Staph group.** Forced vital capacity = FVC; forced expiratory volume in one second = FVE. Bars represent the average of the air volume, (**) means significant difference among treatments by Mann-Whitney test ($P <0.05$), n = 20 per group.

**Fig 4. Pearson's correlation showing significant correlations (*P* < 0.05) between microbiota and clinical variables in the Staph group.** Asterisks represent significant correlations. The bar in the right shows the correlation type, with positive in blue and negative in red. A positive correlation means that two variables in the matrix increased, n = 20.

group, the Staph group, had higher abundance of Staph and *S. aureus*, and the second group, the Pseud group, had higher abundance of Pseud and *P. aeruginosa* (**Fig 5**). In addition, vectors in the plot representing the clinical variables of patients were in the opposite direction to the Staph group, showing an inverse correlation of these parameters with bacteria present in high quantities in this group (**Figs 4** and **5**).

## Prediction of bacterial functional profiles

Functional profiles were predicted from the 16S rRNA data obtained using the software package Tax4Fun. The aim of this analysis was to highlight the different profiles among pathogens groups in an unbiased manner. The complete profiles are shown in the Supporting Information (**S2 Table**). Genes encoding key enzymes involved in virulence pathways were identified in the resulting profiles using their KEGG orthologs (**Table 1**). Thus, key genes related to antibiotic resistance were identified in the Staph group, such as: K01467, beta-lactamase; K03327, multidrug resistance protein, MATE family; K08218, MFS transporter-PAT family, beta-lactamase induction signal transducer AmpG, and genes related to horizontal gene transfer, such as: K07481, transposase, IS5 family; K07485, transposase; and K07489, transposase. In the Pseud group, key genes related to secretion systems were: K02456, general secretion pathway protein G; K02459, general secretion pathway protein J; K03195, type IV secretion system protein VirB10; K07344, type IV secretion system protein TrbL; K11891, type VI secretion system protein ImpL; K11896, type VI secretion system protein ImpG; and a gene related to biofilm synthesis: K11937, biofilm PGA synthesis protein PgaD (**Table 1**).

## Discussion

The bacterial taxa detected in the present study are in agreement with previous studies [27, 28]. Khanolkar *et al.* [29] and Raghuvanshi *et al.* [30] showed shifts in the composition of

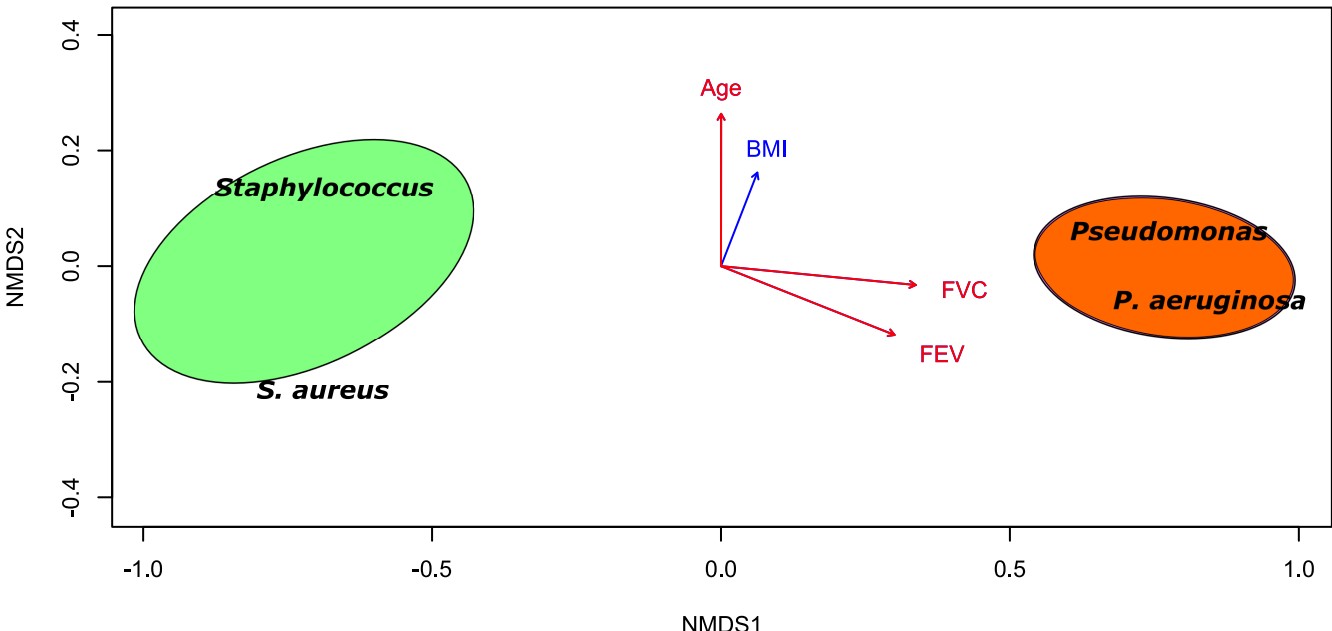

**Fig 5. Non-metric Multidimensional Scaling (NMDS) plot showing significant correlations ($P < 0.05$) between microbiota and clinical variables in the Staph group.** The ellipses encompass different groups, with the Staph group in green and Pseud group in orange. Taxa are shown in black and clinical variables in red and blue vectors (red means significant). The direction of the vectors FEV and FVC indicates an inverse relationship with bacteria present in high quantities in the Staph group, n = 20 per group.

respiratory microbiota in patients with CF, such as the enrichment of *Staphylococcus* spp., *Haemophilus* spp., *Pseudomonas* spp., *Streptococcus* spp., *Serratia* spp., *Neisseria* spp., and *Porphyromonas* spp. We classified patients into two groups based on the dominant pathogens observed in the respiratory microbiota: the Staph and Pseud groups. These taxa are consistent with the results obtained from microbiological cultures, clinical practice, and scientific articles on CF, especially in our target age range [31]. In older patients, other bacteria, such as *Burkholderia* spp., and some emerging bacteria, such as *Stenotrophomonas* spp. and *Acinetobacter* spp. are observed [32].

In Brazil, the diagnosis of CF follows the Brazilian guidelines for the diagnosis and treatment of cystic fibrosis [33]. Thus, the algorithm of newborn screening for cystic fibrosis used in Brazil is based on two tests of immunoreactive trypsinogen levels, the second of which is performed within 30 days of life. If screening is positive (i.e., two positive tests), sweat testing is performed to confirm or rule out cystic fibrosis. Sweat chloride concentrations $\geq$ 60 mmol/ L, as measured by quantitative methods, in two samples, confirm the diagnosis. Diagnostic alternatives are detection of two cystic fibrosis-related mutations and CFTR functional tests.

Children in the Staph group showed lower BMI than the Pseud group (18.09 versus 19.41); that is, children in the Staph group were underweight. Overall, children and young patients infected with *Pseudomonas* usually have a lower BMI than children infected with genus *Staphylococcus*, but this does not occur when the patient is already being monitored and receiving medication; in Pseud patients, treatment includes chronic use of inhaled antibiotics, such as tobramicyn (TOBI®) and azithromycin, which have immunomodulatory and antiviral effects [34]. However, it is not common to use prophylactic treatment in patients with chronic infection by *Staphylococcus* spp.; those patients use "off label" antibiotics [35]. This could explain

**Table 1. Prediction of bacterial functional profiles.**

| KEGG functions |
| --- |
| **Staphylococcus group** |
| K01467; beta-lactamase |
| K02028; polar amino acid transport system ATP-binding protein |
| K02029; polar amino acid transport system permease protein |
| K02030; polar amino acid transport system substrate-binding protein |
| K03327; multidrug resistance protein, MATE family |
| K06994; putative drug exporter of the RND superfamily |
| K07481; transposase, IS5 family |
| K07485; transposase |
| K07489; transposase |
| K07668; two-component system, OmpR family, response regulator VicR |
| K08138; MFS transporter, SP family, xylose: H+ symportor |
| K08191; MFS transporter, ACS family, hexuronate transporter |
| K08218; MFS transporter, PAT family, beta-lactamase induction signal transducer AmpG |
| K11068; hemolysin III |
| K11070; spermidine/putrescine transport system permease protein |
| K11071; spermidine/putrescine transport system permease protein |
| K15342; CRISP-associated protein Cas1 |
| **Pseudomonas group** |
| K02456; general secretion pathway protein G |
| K02459; general secretion pathway protein J |
| K02657; twitching motility two-component system response regulator PilG |
| K03195; type IV secretion system protein VirB10 |
| K03808; paraquat-inducible protein A |
| K07344; type IV secretion system protein TrbL |
| K11891; type VI secretion system protein ImpL |
| K11896; type VI secretion system protein ImpG |
| K11937; biofilm PGA synthesis protein PgaD |
| K12516; putative surface-exposed virulence protein |
| K13735; adhesin/invasin |

the decreased BMI in patients with *Staphylococcus* spp.; however, this information should be used with caution because *S. aureus* is more prevalent at an earlier age and there are no reliable tools to measure lung function in children under six years of age.

Our results showed that patients' clinical condition agreed with the results of pulmonary function (FVC and $FEV_1$). In clinical practice, this observation reflects the definition of disease exacerbation, which means worsening of symptoms, changes in sputum color, loss or cessation of weight gain, and worsening of lung function [10]. Greater patient numbers were observed in the Staph exacerbation group than in the Pseud group, a finding that was confirmed by the lower values of BMI and pulmonary function. Limoli *et al.* [36] observed that co-infection with *S. aureus* and *P. aeruginosa* was associated with decreased lung function and increased numbers of pulmonary exacerbations. Polymicrobial dynamics may be a better indicator of CF patient outcomes, as opposed to the presence of a single pathogen [37, 38]. Pearson's correlation analysis revealed significant relationships between the Staph group and parameters of pulmonary function (FVC and FEV).

In this study, biofilm formation capacity was observed in *S. aureus* and *P. aeruginosa* isolates. Thus, there is increasing evidence that biofilm-mediated infections facilitate the

development of chronic infectious diseases and recurrent infections [39]. Biofilms are often considered a survival strategy for bacteria, which are facilitated by numerous factors in CF lungs, including mucus accumulation [11]. Previous studies have suggested that antibiotic resistance of bacteria in CF lungs is due to biofilm formation [40]. In addition, multiple species of lung biofilm producers such as Pseud in CF patients are affected by specific treatments; thus, competitiveness among different species is harmful, promoting the survival of the most abundant pathogen [41]. However, the clinical significance of *in vitro* biofilm production remains unclear and biofilm detection by laboratory techniques does not necessarily indicate *in vivo* production because biofilms are a community of multiple bacterial species that coexist in a specific environment [42].

Functional inference of communities showed that the presence of key genes in each pathogen group was possible because of the low biodiversity of each group, which was dominated by a single bacterial genus (Staph or Pseud). Thus, in the Staph group, a functional profile determined by antimicrobial resistance genes was observed. In the case of our isolates, this resistance profile was not identified, and all *S. aureus* isolates were sensitive to oxacillin and vancomycin. As in Voronina *et al*. [32], the presence of the *mecA* gene in sputum samples from pediatric patients with CF was not identified in this study; the *mecA* gene confers resistance to methicillin in *S. aureus* strains. The *in silico* inference profile is based on genomes deposited in the database; thus, these genomes may represent strains that carry genes of antibiotic resistance, and the result will depend on the database used as a comparison [43]. In the Pseud group, a dominant profile by secretion systems was identified which is expected because gram-negative bacteria carry several of these systems [44].

There are some limitations in this study. First, the cohort size is too small and heterogeneous. However, it represents the largest pediatric hospital in Brazil, and thus is an interesting clinical cohort from Brazil. In addition, this work is novel, as the only published work on the respiratory microbiota of Brazilian CF patients was recently published by Vasco *et al*. [45], where the authors evaluated the microbiota of 10 children under 6 years old with pancreatic insufficiency who underwent pancreatic enzyme replacement therapy with Creon®. In this sense, this preliminary work is totally different from ours.

The second limitation is regarding the over-simplification of the microbiota data. Initially, we used longitudinal data and triplicates, but this information was not used at all in the manuscript. This information would be relevant to answer important questions such as the longitudinal relationship between microbiome and lung function or the heterogeneity of sputum at a single timepoint. However, analyses of microbiota by the time of collection or by related clinical variables did not identify any specific microbiota pattern from the respiratory tract of the patients. Thus, considering this result, the microbiota was clustered into two main groups, Staph and Pseud. Microbiota samples with abundance >50% of either *Staphylococcus* or *Pseudomonas* were further analyzed. This grouping parameter may seem to skew the data to the expected result. However, this approach, using a cutoff in the data, yielded results that had not been observed in previous studies. Using a cutoff in the data allows the creation of a reference for future studies, and, importantly, this approach helped us to better understand the relationship between opportunistic microbiota pathogens and lung function.

Finally, in patients with CF, the composition of respiratory microbiota varies noticeably between individuals; some patients show marked changes in the bacterial community with alternating infectious agents. Besides, different types of CF mutation did not show unique microbiota. Thus, some points require further research: (i) How can we promptly identify the agent in patients with acute exacerbation with negative classical culture? (ii) How can we make better use of next-generation sequencing and other techniques to identify low-abundance microorganisms that are likely to be responsible for exacerbation in patients?

## Supporting information

**S1 Table. Clinical variables of the patients enrolled in this study.**
(XLSX)

**S2 Table. Complete prediction profiles.**
(XLSX)

**S1 Fig. Relative frequency of the most abundant genera of the bacterial community by data collection.** (A) day 0; (B) day 42; and (C) day 84 of collection.
(EPS)

**S2 Fig. Venn diagram representing the core microbiota of the community by sampling time.** 155 taxa were identified as the core microbiota.
(EPS)

## Acknowledgments

We thank GoGenetic for amplicon sequencing. We would like to thank Editage (www.editage. com) for English language editing.

## Author Contributions

**Conceptualization:** Paulo Kussek, Helisson Faoro, Jussara Kasuko Palmeiro, Libera Maria Dalla Costa.

**Data curation:** Dany Mesa, Humberto Ibanez.

**Formal analysis:** Dany Mesa, Damaris Krul.

**Funding acquisition:** Libera Maria Dalla Costa.

**Investigation:** Damaris Krul.

**Methodology:** Thaís Muniz Vasconcelos, Luiza Souza Rodrigues.

**Project administration:** Libera Maria Dalla Costa.

**Writing – original draft:** Dany Mesa.

**Writing – review & editing:** Dany Mesa, Luiza Souza Rodrigues, Helisson Faoro, Jussara Kasuko Palmeiro, Libera Maria Dalla Costa.

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
