## [Decision Letter · Decision Letter 0]

24 Mar 2022

PONE-D-22-04509Upper airway microbiota and decreasing lung function in young cystic fibrosis brazilian patients with pulmonary Staphylococcus and Pseudomonas infectionPLOS ONE

Dear Dr. Dany Mesa,

Thank you for submitting your manuscript to PLOS ONE. After careful consideration, we feel that it has merit but does not fully meet PLOS ONE’s publication criteria as it currently stands. Therefore, we invite you to submit a revised version of the manuscript that addresses the points raised during the review process.

We look forward to receiving your revised manuscript.

Kind regards,

Abdelwahab Omri, Pharm B, Ph.D, Laurentian University

Academic Editor

PLOS ONE

Journal Requirements:

2. At PRTC, please send back with the following note.

"Please provide additional details regarding participant consent. In the ethics statement in the Methods and online submission information, please ensure that you have specified what type you obtained (for instance, written or verbal, and if verbal, how it was documented and witnessed). If your study included minors, state whether you obtained consent from parents or guardians. If the need for consent was waived by the ethics committee, please include this information.

Thank you for stating the following financial disclosure: 

"No"

Reviewers' comments:

Reviewer's Responses to Questions

**Comments to the Author**

1. Is the manuscript technically sound, and do the data support the conclusions?

Reviewer #1: Partly

Reviewer #2: Yes

Reviewer #3: Yes

Reviewer #4: Partly

2. Has the statistical analysis been performed appropriately and rigorously? 

Reviewer #1: No

Reviewer #2: Yes

Reviewer #3: Yes

Reviewer #4: Yes

3. Have the authors made all data underlying the findings in their manuscript fully available?

Reviewer #1: Yes

Reviewer #2: Yes

Reviewer #3: Yes

Reviewer #4: No

4. Is the manuscript presented in an intelligible fashion and written in standard English?

Reviewer #1: Yes

Reviewer #2: Yes

Reviewer #3: Yes

Reviewer #4: Yes

5. Review Comments to the Author

Reviewer #1: Mesa et al present a longitudinal prospective study of predominantly pediatric CF patients to evaluate their microbiome and correlate it to clinical findings. While this is certainly an area of interest, a number of significant issues preclude my recommendation for publication as noted in detail below.

Title: I am very unclear why it states upper airway microbiota when the authors clearly describe induced sputum which samples mainly the lower respiratory tract. While there have been numerous studies evaluating the presence of upper airway oropharyngeal microbiota contamination in saliva samples--to label this as primarily upper airway (i.e. oropharynx and proximal anatomy) is incorrect and atypical in CF literature.

Abstract:

Line 26: "Long periods"--this is vague--are they referring to clinical progression? Baseline?

Line 29-30: This is inaccurate to say PEx are caused by those classic pathogens--while that has been the traditional viewpoint, the authors should be redirected to numerous studies failing to demonstrate molecular (i.e. increased relative abundance) or culture-based (i.e. CFU) evidence that classic pathogens increase at times of PEx--hence this is the cornerstone of microbiome studies in CF.

Line 32: Need a space between "time to".

Line 33: Again not upper respiratory tract..

Line 33: Unclear what "more abundant" pathogens refers to--by prevalence or relative abundance in samples?

Line 37: Had should be lower case.

Line 39: Unclear what the patients clinical condition were consistent with results of pulmonary function?

Introduction:

Line 52: This is incorrect, delF508 is present in >90% of CF patients in one copy--unless the authors are describing homozygous delF508 which then should be included in the sentence.

Line 54: "A marked inflammatory process" suggests only one process which is inaccurate.

Line 55: I am unclear what "pulmonary lesions" are. ?Bronchiectasis? These are not lesions.

Line 57: This needs a reference as up until a decade ago the respiratory tract was considered sterile.

Line 67-68: Disagree. In fact numerous taxa have been implicated in CF as both pathogenic or protective--if the authors are going to make such a sweeping statement a reference is required.

Lines 76-78: Again refer to the same statement in the abstract which is inaccurate.

Lines 83-84: Poor sentence structure, they describe their study but also present conclusions.

Methods:

Lines 108-115: The authors should reference where they obtained these definitions of baseline, exacerbation, recovery and treatment.

Lines 119: Fragment sentence.

Lines 120: More discussion around the control group in the study by Hahn et al is required in the methods briefly--including what defined "healthy" and if any exclusion criteria were applied.

Line 132: This is unclear--where sputum collected in triplicate on each of those days (0, 42, 84) OR was it one sample on each of these days?

Results:

Line 216: Can the authors comment more (perhaps in the methods) on CF diagnosis in Brazil? In North America this done by newborn screening but the authors imply that 31 had an early diagnosis before age 2. More clarity would be helpful to international audiences.

Lines 220: Define eutrophic, unclear what this and grades of malnutrition are.

Line 224: Should put percentage of patients next to number.

Lines 226: Are "both organisms in association" on the SAME culture or just a history of both?

Line 233: I am assuming the authors mean relative abundance but this should be clarified.

Lines 235: The authors need to expand on these results--was there assay a dichotomous yes/no or were there grades of biofilm development that could be scored?

Lines 238-239: This is very well known in the literature and not required in the results.

Lines 243: Are these ASV's since the authors used the dada2 protocol? If so they should consider presenting their data at the collapsed genera level such to not split ASV (i.e. two Prevotella are listed).

Lines 247-249: This is very much lacking in detail. How was this analysis done? Was it a multivariate model? Where are the statistics or statistical methodology?

Lines 250-252: Again, unclear is. Is it they had to be BOTH culture AND most abundant or either or? I also find it suspect that each had 20 samples per group--more clarity is required. Why did the authors pick Pseudomonas and Staphylococcus? They stated that the most abundant genera were mostly anaerobic--so what is the rationale for this? If it is to evaluate classic pathogens, why exclude Haemophilus which appears to be prevalent and abundant? I do not see a clear hypothesis.

Lines 265: The authors need to expand how they got to the species level, again by ASV?

Lines 281-285: This is absolutely incorrect. The authors have not shown any statistical analysis to evaluate for confounders before concluding that Staphylococcus is contributing to underweight status. There is lacking of evidence to back up such a sweeping conclusion.

Lines 286-287: Is this nebulizer chronic therapy?? Again more details.

Lines 296-298: I find it unusual their Pseudomonas group had BETTER clinical outcome when multiple studies over years clearly show worse outcomes upon acquisition of Pseudomonas. The authors need to postulate in the discussion why their results differ from multiple well-conducted studies.

Lines 316-332: Again I am unclear what this data is meant to show. It is clear the dominant groups would differ--what is the hypothesis?

Discussion:

Overall, I found the discussion weak and unable to support many of their conclusions for the reasons cited in the results. Specific examples include: lines 377-388 with no discussion around confounding variables.

Many of the references are not appropriate or up to date to support the conclusions stated--such as reference 32. Overall, I do not see a clear hypothesis or understanding of how the data contributes to the literature. It was also extremely unclear why the authors described a longitudinal prospective study but then did no work on this and rather became cross-sectional.

Overall significant issues are present and needs careful attention.

Reviewer #2: In the study presented here, authors Kussek et al. show data on the respiratory microbiota of children with cystic fibrosis in Brazil. 34 children with CF are included in the study, and samples are collected at 3 different time points for comparison. Relationships between clinical variables (lung function scores, BMI, etc.) and abundance of dominant CF respiratory pathogens, namely S. aureus and P. aeruginosa, are investigated. The authors conclude that S. aureus correlates with worse clinical outcomes in their pediatric CF population, and they identify potential virulence genes in S. aureus and P. aeruginosa metagenomic data.

The data presented on this population of children with CF is valuable and addresses important questions about how presence of CF bacterial pathogens relates to patient exacerbations and clinical outcomes. The manuscript is overall well-written, and the strengths and weaknesses of this work is well described in the discussion section. This would be a valuable dataset to publish, if concerns raised below are addressed:

Major concerns:

A major issue with this work is that healthy control data used for comparison is from a previous publication by a different group who conducted their study in the US (lines 115-121). Samples from healthy children in the Hahn et al. study were collected from a different site in the respiratory tract (oropharynx) than used in this manuscript, and data were obtained by sequencing samples from oropharyngeal swabs, not induced sputum samples. OP swabs have not been reported to accurately reflect the lower respiratory microbiome, and there may also be regional differences in respiratory microbiota in healthy controls in distinct geographic locations like the US and Brazil. For these reasons, making direct comparisons between these datasets is not possible, and healthy control data should be removed from figures and tables presented here. The new data collected for this study on the microbiome of children in Brazil with CF is valuable on its own. It would be appropriate compare and contrast how these findings relate to previous studies with healthy control groups of a similar age in the discussion section, but not in the results.

It would be informative to compare the S. aureus- and P. aeruginosa-dominant groups to samples where neither of these organisms was the abundant species in Fig. 2 and Fig. 3, rather than making the healthy control comparisons using data from the Hahn et al. study.

For analyses in Fig. 2, 20 samples per group with greater than 50% relative abundance of S. aureus and P. aeruginosa were selected. If not all samples meeting this criteria were compared, how were representative samples chosen? Are samples evaluated here from the same patients whose clinical variables (FEV and FVC) are compared in Figure 4?

The NMDS plot in Fig. 6 does not show individual data points for samples, so interpreting how ellipses relate to clustering of the different groups is not possible. It is confusing how four clinical variables can be included here in a plot with only two axes. It would seem correct to instead separate the clinical variables into different plots where 3 variables are compared in each and show multiple plots to make comparisons.

Minor concerns:

Lines 29-30 and 74-77, There is not a concrete link between either P. aeruginosa or S. aureus and exacerbation in CF, although many groups have looked at this and presence of bacteria has been positively correlated with other outcomes (i.e. P. aeruginosa infection and end-stage lung disease). Would consider walking back language in these statements.

Lines 32-33, The data presented is from induced sputum samples obtained by inhalation of nebulized hypertonic saline, which would provide information about lower respiratory tract infections and microbiota. However, the abstract states that upper respiratory microbiota are evaluated, which is not tested here. This should be changed to “lower respiratory microbiota.”

Line 53, Rather than stating “CF beings early in life…” it may be more accurate to say “Complications of CF disease begin early in life…”

Line 60, Does “its mutation class” refer to CFTR mutations?

Line 64, Would remove the word “all” from statement that “… all pulmonary distal airways are inaccessible.”

Lines 244-245, How is the “core community” defined? Based on Figure S3, would this be taxa identified at all sampling periods, and/or in all samples?

Figure 4, It would be helpful to include individual data points for comparison of sample groups in bar graphs.

Lines 286-287, Was it evaluated if receiving antimicrobial treatment led to significant changes in P. aeruginosa relative abundance at subjects’ next visits post-treatment? This would be a very interesting relationship to investigate.

Lines 440-453, It’s discussed that other comparisons were made with longitudinal samples, but data was not included because a pattern was not identified. I would consider revising or removing some of this language; not all data is going to result in a pattern, and stating that a particular outcome is sought indicates potential bias in data interpretation. Variability in patient clinical data is common and expected. These findings actually echo much of what has been observed in other CF patient studies, and presenting results even if a specific correlation isn’t observed is still valuable.

Reviewer #3: The manuscript describes a study undertaken to assess the upper respiratory tract microbiota of young patients suffering from cystic fibrosis (CF). Sputum samples collected from 34 CF patients were characterised and further used for isolating microbial cultures. Parallelly, the samples were also used to extract the DNA and perform 16SrRNA gene sequencing to analyse the entire microbial community present. Functional profiles of genes encoding virulence factors or antibiotic resistance were analysed and used for correlation with microbial data.

The manuscript is well written and the authors have rightly described the few limitations of the study also highlighting the need to use next generation sequencing and other techniques to better assess the data.

After thoroughly reading the article, I would like to make the following comments and suggestions.

1] Abstract: appears a bit general. It would be appropriate to include specific results obtained and conclusions drawn. Correct the typographical errors on Line 32 and Line 37.

2] Introduction is well written and identifies the need to carry out this study. Full name (Genus and species) of organism may be written on first appearance followed by genus (initial) and species full name.

3] Materials and Methods-

i. Line 161-162- It would have been more appropriate to measure the absorbance to quantify the biofilm synthesised. Visual (Qualitative data) interpretations are less reliable.

ii. Line 163-164 – Please mention the strain numbers or source of Candida isolates used.

4] Results-

i. Line 233- How was the abundance of a particular pathogen determined?

ii. Line 243- Prevotella 7? Appears to be a typographical error

iii. Line 300- Figure 4A and 4B can be merged

iv. Line 341- How reliable is to predict the bacterial functional profiles on the basis of 16SrRNA gene sequence data alone? Justify.

v. Were only 23 (13 S. aureus and 10 P. aeruginosa) isolates obtained on culturing sputum samples? Or was that the number tested for biofilm formation? Did you isolate both S. aureus and P. aeruginosa both from a single sputum sample?

vi. The term Staphylococcus aureus and Staphylococcus spp. appears to be used quite interchangeably. Please correct the same. Same for Pseudomonas aeruginosa and Pseudomonas spp.

5] Discussion is well written.

6] There are few grammatical and typographical mistakes in the manuscript which need to be carefully identified and corrected.

Reviewer #4: General Comments:

It is an important non-interventional clinical trial in which sputum samples of young patients were evaluated for the relative prevalence of bacteria in their sputum. There are a lot if information collected however the way the data are presented makes the study findings and conclusions not clear and the significance of the findings not convincing. In fact, discussion of the results is stressing that the data generated simply confirms previously published studies.

The analysis of the results needs a major revision after the patients are properly grouped according to their mutation status, sex and baseline FeV1. The results generated should be re-evaluated once the patients are properly divided according to their mutation status and their exacerbation status at the onset of the trial. The exacerbating patients can not be analyzed in the same group as stable patients.

The authors state that FEV1 of 26% for some patients. It is unclear if this is a baseline FeV1 typical for the patients or recorded at the peak of exacerbation. If is a baseline FeV1 it would be exceptionally severe CF lung disease for so young patient (s). If this only happened for few days at the peak of exacerbation when patient experienced severe pneumonia it is completely different story. The FEV1 at onset should be measured when the patient is stable for 30 days.

FeV1 can be measured also during the exacerbation of course to evaluate how severe impairment in lung function occurred during the exacerbation but the analyses of the findings should be done by separating the patients into different subgroups groups and interpretation of these findings should to be different when the stable patients are evaluated vs. when the exacerbating patients are evaluated.

The exacerbating patients’ microbiota in the sputum should be evaluated separately than microbiota typical for steady state baseline microbiota typical for chronically infected patient on azithromycin and inhaled antibiotics.

The way the data are presented does not brings much useful information. It is not clear what is a typical the microbiome at steady-state level in each patient and which microbiome changes occur during exacerbation .

Is there observed during exacerbations an increase in the CFUs of specific pathogens that were already present at baseline (with the relative composition of bacteria remaining the same), or during the exacerbation the relative composition of bacteria changes dramatically?

Major specific comments:

1. Patients ages ranges 8-23 contain both pediatric cases and adult cases so should not be classified broadly as pediatric. Table containing characteristics of each patient should be included that would provide information about each patient’s age, sex, the class of the mutations that each patient had should also be stated, specific CFTR gene mutations should be stated for each patient, CRP at the time of the initiation of the trial ( for stable patients with no exacerbation for 30 days), amount of exacerbation in the year prior to the study at the initiation of the study. Steady-state FeV1 recorded when there is no exacerbation.

2. FEV1 is only informative as a characteristic of the patient when is recorded at baseline (30 days with no exacerbation and with not major increase in CRP level compared to the previous periods without exacerbation). The patients with active exacerbation at the onset of the study should be studied separately and once they get stabilized their baseline Fev1 should be recorded. At the exacerbation the FeV1 is very volatile and can drop dramatically but it does not represent the overall FeV1 characteristics of the patient. It is affected primarily by the sudden raise in the bacterial counts and sometimes also type of combination of infections

3. The biggest drop in the stable FEV1 usually occurs between age 18 and 30. It is not clear how many patients were in this age bracket? Did the authors seen a difference in baseline FeV1 and between 8–17-year-old patients compared to 18–23-year-old patients?

4. The number of patients positive for PA, SA and both should be stated, not only the percentage of the cohort L225-L228. How many patients experienced dual infection with PA and SA? (Table S1…. check if the information is there….)

5. Is the relative higher abundance PA vs SA in both age group similar?

6. Methodology mentioned in 146-149 needs to be properly described in detail (buffers, duration of run, columns used, temperature, elution parameters, etc … ). The methodology should be written in a way that the method can be reproduced by other investigators just based on the descriptions provided. The protocol used should be included as a supplementary material. If there as space restrictions imposed by the journal, detailed protocols, including details, buffers, parameters of runs, how many times repeated, standards used should be appended as supplemental data. The way it is written is not informative at all and can not be even properly evaluated.

7. Primers sequences should be included 5’-3’ in addition to their positions that has been mentioned in L178.

8. The information about CFTR gene mutation presented as percentages should be at least referred to the Table in the supplement that contains specific information regarding CFTR mutations each patient included in the study had, so not only percentages would be presented but the number of patients for each mutation type combination should be clear

9. The description of BMI does not take into consideration if the patients were males or females the range of BMI changes with age and sex, some children grow faster than other children so the way the BMI is presented in not informative at all.

10. In the discussion, the authors state that the Staphylococcal infections are not usually treated because there is not good treatment protocol. Authors findings suggest that the Staphylococcus species tested during this study were sensitive to oxacillin and vancomycin. It is not clear if these laboratory findings were used in the study to select proper treatment for Staphylococcal exacerbation or not. This should be at least mentioned.

11. The way the authors group the data into two groups over 50% Staphylococcus spp. or Pseudomonas spp. made the analysis of the data and resulting conclusions predictable and expected and diminished clinical significance of presented findings.

12. Lack of identification of specific microbe associated with apparent exacerbation is not well discussed; possibility of viral infections (frequently not diagnosed because there is no specific diagnostic test established, or/and the physician does not asks for more sophisticated tests to be done) and very slow growing or difficult to propagate in vitro bacterial infections (e.g mycobacterial species, such as M. abscessus complex)

Minor editing errors:

Abstract: Sentence starting from Clustering…. and the sentence starting from “Had reduced….” Should be all one sentence

L52: Authors state the 48% patients were delF508 – this is not clear – 48% had one allele of delF508 or both alleles with the same delF508 mutation?

6. PLOS authors have the option to publish the peer review history of their article (what does this mean?). If published, this will include your full peer review and any attached files.

Reviewer #1: No

Reviewer #2: No

Reviewer #3: No

Reviewer #4: No

---

## [Author Response · Author response to Decision Letter 0]

5 Aug 2022

June 23, 2022

Abdelwahab Omri, Pharm B, Ph.D

Academic Editor

PLOS ONE

Dear Editor:

I, along with my coauthors, wish to submit the revised manuscript entitled “Lower airway microbiota and decreasing lung function in young cystic fibrosis Brazilian patients with pulmonary Staphylococcus and Pseudomonas infection.” The paper was coauthored by Paulo Kussek, Dany Mesa, Luiza Souza Rodrigues, Thaís Muniz Vasconcelos, Damaris Krul, Helisson Faoro, Humberto Ibanez, and Jussara Kasuko Palmeiro.

We would like to thank you and the reviewers for your thoughtful suggestions and insights which have significantly helped us to improve our manuscript. We have carefully revised the manuscript (the revisions are highlighted in yellow for your convenience) to address the major concerns raised and hope that our revised manuscript meets your standards and will be reconsidered for publication in PLOS ONE. 

We have also provided our point-by-point response (in red color) to all comments raised and hope that our replay addressed all your concerns.

Should you have any further questions, please contact us.

Thank you for your consideration. I look forward to hearing from you. 

Sincerely,

Libera Maria Dalla Costa

Faculdades Pequeno Príncipe (FPP), Curitiba, Paraná, Brazil

Instituto de Pesquisa Pelé Pequeno Príncipe (IPPPP), Curitiba, Paraná, Brazil

Av. Silva Jardim, 1632, Curitiba, PR, Brazil

Fax: +55 41 33101035 

lmdallacosta@gmail.com

libera.costa@professor.edu.br

 

Specific comments 

Reviewer #1: 

Mesa et al present a longitudinal prospective study of predominantly pediatric CF patients to evaluate their microbiome and correlate it to clinical findings. While this is certainly an area of interest, a number of significant issues preclude my recommendation for publication as noted in detail below.

Thank you for your constructive suggestions. We modified the structure to make the text clearer from the introduction to the discussion. In addition, each question was answered in red color.

Title: I am very unclear why it states upper airway microbiota when the authors clearly describe induced sputum which samples mainly the lower respiratory tract. While there have been numerous studies evaluating the presence of upper airway oropharyngeal microbiota contamination in saliva samples--to label this as primarily upper airway (i.e. oropharynx and proximal anatomy) is incorrect and atypical in CF literature. 

Thank you for your observation. The title was corrected. Sorry for the mistake.

Abstract:

Line 26: "Long periods"--this is vague--are they referring to clinical progression? Baseline? 

Yes, this refers to the clinical progression of the disease. The text was modified to make the statement clearer.

Line 29-30: This is inaccurate to say PEx are caused by those classic pathogens--while that has been the traditional viewpoint, the authors should be redirected to numerous studies failing to demonstrate molecular (i.e. increased relative abundance) or culture-based (i.e. CFU) evidence that classic pathogens increase at times of PEx--hence this is the cornerstone of microbiome studies in CF. 

Thanks for your observation. The sentence has been reworded.

Line 32: Need a space between "time to". Thank you, space added.

Line 33: Again not upper respiratory tract. Thank you. The sentence was corrected.

Line 33: Unclear what "more abundant" pathogens refers to--by prevalence or relative abundance in samples? Thank you, the sentence was adjusted.

Line 37: Had should be lower case. Thank you, capital letter replaced.

Line 39: Unclear what the patients clinical condition were consistent with results of pulmonary function? Thank you, the patient's clinical condition was to make the sentence clearer.

Introduction:

Line 52: This is incorrect, delF508 is present in >90% of CF patients in one copy--unless the authors are describing homozygous delF508 which then should be included in the sentence. Thanks you, the statement was corrected in line 53.

Line 54: "A marked inflammatory process" suggests only one process which is inaccurate. Thanks you, the statement was adjusted in lines 55 and 56.

Line 55: I am unclear what "pulmonary lesions" are. ?Bronchiectasis? These are not lesions. Thank you, the sentence was adjusted in line 56.

Line 57: This needs a reference as up until a decade ago the respiratory tract was considered sterile. Thank you, a reference was included in line 59.

Line 67-68: Disagree. In fact numerous taxa have been implicated in CF as both pathogenic or protective--if the authors are going to make such a sweeping statement a reference is required. Thank you, the sentence was adjusted in lines 68 and 69.

Lines 76-78: Again refer to the same statement in the abstract which is inaccurate. Thank you, the sentence was adjusted in line 76.

Lines 83-84: Poor sentence structure, they describe their study but also present conclusions. Thank you, the sentence was adjusted.

Methods:

Lines 108-115: The authors should reference where they obtained these definitions of baseline, exacerbation, recovery and treatment. Thank you, a reference was included in line 107.

Lines 119: Fragment sentence. Thank you, the sentence was excluded.

Lines 120: More discussion around the control group in the study by Hahn et al is required in the methods briefly--including what defined "healthy" and if any exclusion criteria were applied. Thank you for the suggestion, however the control group was excluded.

Line 132: This is unclear--where sputum collected in triplicate on each of those days (0, 42, 84) OR was it one sample on each of these days? Thank you, the sentence was adjusted in line 124.

Results:

Line 216: Can the authors comment more (perhaps in the methods) on CF diagnosis in Brazil? In North America this done by newborn screening, but the authors imply that 31 had an early diagnosis before age 2. More clarity would be helpful to international audiences. Patients were confirmed with CF by a sweat test and CFTR gene screening (lines 98-99 of methods). In addition, one sentence regarding diagnosis in Brazil was added in discussion section (Line 360-367).

Lines 220: Define eutrophic, unclear what this and grades of malnutrition are. Anthropometry is a useful tool for grading malnutrition (reference 1) and dates demonstration of the correlation between severity of underweight for age and risk of death (more pulmonary exacerbations and major risk of hospitalization (reference 2). The publication by WHO opened the way to validation of different indicators in terms of their ability to predict risk of death in different geographical or emergency situations with reference to a globally accepted database (reference 3). References were added to the text in lines 216 and 217.

1. Duggan MB. Anthropometry as a tool for measuring malnutrition: impact of the new WHO growth standards and reference. Ann Trop Paediatr. 2010;30(1):1-17. doi: 10.1179/146532810X12637745451834. PMID: 20196929.

2. Smyth AR, Bell SC, Bojcin S, Bryon M, Duff A, Flume P, Kashirskaya N, Munck A, Ratjen F, Schwarzenberg SJ, Sermet-Gaudelus I, Southern KW, Taccetti G, Ullrich G, Wolfe S; European Cystic Fibrosis Society. European Cystic Fibrosis Society Standards of Care: Best Practice guidelines. J Cyst Fibros. 2014 May;13 Suppl 1:S23-42. doi: 10.1016/j.jcf.2014.03.010. PMID: 24856775.

3. Weir CB, Jan A. BMI Classification Percentile And Cut Off Points. 2021 Jun 29. In: StatPearls [Internet]. Treasure Island (FL): StatPearls Publishing; 2022 Jan–. PMID: 31082114.

Line 224: Should put percentage of patients next to number. Thank you for your suggestion. Percentage was added in lines 221-223.

Lines 226: Are "both organisms in association" on the SAME culture or just a history of both? Both organisms were isolated from the same culture sample but in different culture media.

Line 233: I am assuming the authors mean relative abundance but this should be clarified. Thank you, the sentence was corrected.

Lines 235: The authors need to expand on these results--was there assay a dichotomous yes/no or were there grades of biofilm development that could be scored? Only the dichotomous analysis, presence or absence, was used, considering that by visual analysis the interpretation of intensity (+/++ and +++) can be subjective, especially without specific controls for each of these categories. Lines 232-234.

Lines 238-239: This is very well known in the literature and not required in the results. Thank you for your suggestion, the sentence was excluded.

Lines 243: Are these ASV's since the authors used the dada2 protocol? If so they should consider presenting their data at the collapsed genera level such to not split ASV (i.e. two Prevotella are listed) Thank you for your suggestion, the sentence was adjusted and only genera was listed.

Lines 247-249: This is very much lacking in detail. How was this analysis done? Was it a multivariate model? Where are the statistics or statistical methodology? Thank you for your suggestion, the sentence was adjusted, and details were included in the section of material and methods.

Lines 250-252: Again, unclear is. Is it they had to be BOTH culture AND most abundant or either or? I also find it suspect that each had 20 samples per group--more clarity is required. Why did the authors pick Pseudomonas and Staphylococcus? They stated that the most abundant genera were mostly anaerobic--so what is the rationale for this? If it is to evaluate classic pathogens, why exclude Haemophilus which appears to be prevalent and abundant? I do not see a clear hypothesis. Thank you, the sentence was adjusted, and details were included in line 249-252.

Lines 265: The authors need to expand how they got to the species level, again by ASV? According to your recommendation we have removed the control group, so, the figure 3 was removed.

Lines 281-285: This is absolutely incorrect. The authors have not shown any statistical analysis to evaluate for confounders before concluding that Staphylococcus is contributing to underweight status. There is lacking of evidence to back up such a sweeping conclusion. Thank you, the sentence was adjusted in line 269-272. 

Lines 286-287: Is this nebulizer chronic therapy?? Again more details. The treatment of lung disease in cystic fibrosis includes nebulization of various medications that are key to improve lung health, and an inhaler system is essential for all patients with cystic fibrosis. All patients in the Pseudomonas group (except one patient) received antimicrobial treatment against Pseudomonas spp, in other words, chronically inhaled antibiotic to reduce bacterial growth and the frequency of exacerbations. Lines 274-275.

Lines 296-298: I find it unusual their Pseudomonas group had BETTER clinical outcome when multiple studies over years clearly show worse outcomes upon acquisition of Pseudomonas. The authors need to postulate in the discussion why their results differ from multiple well-conducted studies. Thank you for suggestion, some points were addressed in discussion section.

Lines 316-332: Again I am unclear what this data is meant to show. It is clear the dominant groups would differ--what is the hypothesis? Our data mean to analyze the clinical variables in function of these two groups. A different approach to traditional works. 

Discussion:

Overall, I found the discussion weak and unable to support many of their conclusions for the reasons cited in the results. Specific examples include: lines 377-388 with no discussion around confounding variables.

Many of the references are not appropriate or up to date to support the conclusions stated--such as reference 32. Overall, I do not see a clear hypothesis or understanding of how the data contributes to the literature. It was also extremely unclear why the authors described a longitudinal prospective study but then did no work on this and rather became cross-sectional.

Overall significant issues are present and needs careful attention.

Thank you for your considerations. Some points were addressed in discussion section. The objective was to carry out a longitudinal prospective study, however the microbiota data were very different between patients. We performed various statistical analyses without any significant result. Therefore, we opted for a different approach, to group patients by the dominant pathogen observed in the microbiota and compare the clinical variables among these two pathogen groups, Staph and Pseud. We agree that the approach is not common in this area; however it was the only option left to take advantage of this data. Thus, this work in not a “longitudinal prospective study”. 

Reviewer #2: 

In the study presented here, authors Kussek et al. show data on the respiratory microbiota of children with cystic fibrosis in Brazil. 34 children with CF are included in the study, and samples are collected at 3 different time points for comparison. Relationships between clinical variables (lung function scores, BMI, etc.) and abundance of dominant CF respiratory pathogens, namely S. aureus and P. aeruginosa, are investigated. The authors conclude that S. aureus correlates with worse clinical outcomes in their pediatric CF population, and they identify potential virulence genes in S. aureus and P. aeruginosa metagenomic data.

The data presented on this population of children with CF is valuable and addresses important questions about how presence of CF bacterial pathogens relates to patient exacerbations and clinical outcomes. The manuscript is overall well-written, and the strengths and weaknesses of this work is well described in the discussion section. This would be a valuable dataset to publish, if concerns raised below are addressed:

Thanks you for your constructive suggestions. Each question was answered in red color.

Major concerns:

A major issue with this work is that healthy control data used for comparison is from a previous publication by a different group who conducted their study in the US (lines 115-121). Samples from healthy children in the Hahn et al. study were collected from a different site in the respiratory tract (oropharynx) than used in this manuscript, and data were obtained by sequencing samples from oropharyngeal swabs, not induced sputum samples. OP swabs have not been reported to accurately reflect the lower respiratory microbiome, and there may also be regional differences in respiratory microbiota in healthy controls in distinct geographic locations like the US and Brazil. For these reasons, making direct comparisons between these datasets is not possible, and healthy control data should be removed from figures and tables presented here. According to your recommendation we have removed the control group, so, the figure 3 was excluded from the manuscript. 

The new data collected for this study on the microbiome of children in Brazil with CF is valuable on its own. It would be appropriate compare and contrast how these findings relate to previous studies with healthy control groups of a similar age in the discussion section, but not in the results. It would be informative to compare the S. aureus- and P. aeruginosa-dominant groups to samples where neither of these organisms was the abundant species in Fig. 2 and Fig. 3, rather than making the healthy control comparisons using data from the Hahn et al. study. According to your recommendation we have modified the figure 2, the healthy control was excluded from the manuscript.

For analyses in Fig. 2, 20 samples per group with greater than 50% relative abundance of S. aureus and P. aeruginosa were selected. If not all samples meeting this criteria were compared, The selection criterion was the minimal number of identified samples by culture (20) in both groups (Staphylococcus and Pseudomonas), how were representative samples chosen? Are samples evaluated here from the same patients whose clinical variables (FEV and FVC) are compared in Figure 4? Yes, they are the same samples compared in figure 4, now figure 3. The value from these variables is shown in Table S1.

The NMDS plot in Fig. 6 does not show individual data points for samples, so interpreting how ellipses relate to clustering of the different groups is not possible. It is confusing how four clinical variables can be included here in a plot with only two axes. It would seem correct to instead separate the clinical variables into different plots where 3 variables are compared in each and show multiple plots to make comparisons. Non-metric Multi-dimensional Scaling (NMDS) is a way to condense information from multidimensional data (multiple variables/species/OTUs), into a 2D representation or ordination. In this ordination, the closer two points are, the more similar the corresponding samples are with respect to the variables that went into making the NMDS plot. In addition, several works used this approach, for example, the work of Tassi et al, 2018 “The biogeochemical vertical structure renders a meromictic volcanic lake a trap for geogenic CO2 (Lake Averno, Italy)” plotted more than 10 variables in a unique figure. Thus, this approach is scientifically validated.

Minor concerns:

Lines 29-30 and 74-77, There is not a concrete link between either P. aeruginosa or S. aureus and exacerbation in CF, although many groups have looked at this and presence of bacteria has been positively correlated with other outcomes (i.e. P. aeruginosa infection and end-stage lung disease). Would consider walking back language in these statements. Lines 32-33, The data presented is from induced sputum samples obtained by inhalation of nebulized hypertonic saline, which would provide information about lower respiratory tract infections and microbiota. However, the abstract states that upper respiratory microbiota are evaluated, which is not tested here. This should be changed to “lower respiratory microbiota.” According to your recommendation we have rewrote this sentence.

Line 53, Rather than stating “CF beings early in life…” it may be more accurate to say “Complications of CF disease begin early in life According to your recommendation we have rewrote this sentence in line 54.

”Line 60, Does “its mutation class” refer to CFTR mutations? Line 64, Would remove the word “all” from statement that “… all pulmonary distal airways are inaccessible.” According to your recommendation we have rewrote this sentence, the word “all” was also removed.

Lines 244-245, How is the “core community” defined? Based on Figure S3, would this be taxa identified at all sampling periods, and/or in all samples? Yes, all sampling periods, and/or in all samples, according to your recommendation we have rewrote this sentence.

Figure 4, It would be helpful to include individual data points for comparison of sample groups in bar graphs. Considering that standard deviation inclusion in the respective bars of each group, we believe that the figure is adequately informative. In addition, the figure was chance a new figure (Figure 3).

Lines 286-287, Was it evaluated if receiving antimicrobial treatment led to significant changes in P. aeruginosa relative abundance at subjects’ next visits post-treatment? This would be a very interesting relationship to investigate. We only have data from three visits, we haven´t got following up post-treatment data.

Lines 440-453, It’s discussed that other comparisons were made with longitudinal samples, but data was not included because a pattern was not identified. I would consider revising or removing some of this language; not all data is going to result in a pattern, and stating that a particular outcome is sought indicates potential bias in data interpretation. Variability in patient clinical data is common and expected. These findings actually echo much of what has been observed in other CF patient studies, and presenting results even if a specific correlation isn’t observed is still valuable. Thank very much for your positive considerations of this data. 

 

Reviewer #3: 

The manuscript describes a study undertaken to assess the upper respiratory tract microbiota of young patients suffering from cystic fibrosis (CF). Sputum samples collected from 34 CF patients were characterised and further used for isolating microbial cultures. Parallelly, the samples were also used to extract the DNA and perform 16SrRNA gene sequencing to analyze the entire microbial community present. Functional profiles of genes encoding virulence factors or antibiotic resistance were analysed and used for correlation with microbial data. The manuscript is well written and the authors have rightly described the few limitations of the study also highlighting the need to use next generation sequencing and other techniques to better assess the data. After thoroughly reading the article, I would like to make the following comments and suggestions. 1] Abstract: appears a bit general. It would be appropriate to include specific results obtained and conclusions drawn. Correct the typographical errors on Line 32 and Line 37. 

Thanks you for your constructive suggestions. We have modified the text to clarity the manuscript from the introduction until discussion. In addition, each question was answered in red color. 

The typographical errors were corrected.

2] Introduction is well written and identifies the need to carry out this study. Full name (Genus and species) of organism may be written on first appearance followed by genus (initial) and species full name. 

Thank you, Genus and species nomenclature were corrected as your recommendation.

3] Materials and Methods-

i. Line 161-162- It would have been more appropriate to measure the absorbance to quantify the biofilm synthesised. Visual (Qualitative data) interpretations are less reliable. 

We agree that the sensitivity of detection by absorbance is higher and that it would allow its quantification, however, the tube method is validated and here, we had a concordant result in controls, duplicate analyses and in the results between different observers. 

ii. Line 163-164 – Please mention the strain numbers or source of Candida isolates used. 

We used clinical isolates of Candida spp. belonging to the collection of microorganisms of our institution, previously characterized as biofilm producers by quantitative and qualitative assessment methods for biofilm growth (microtiter plate assay and tube method). 

4] Results-

i. Line 233- How was the abundance of a particular pathogen determined? 

Please see the rephrased sentence in the manuscript. 

ii. Line 243- Prevotella 7? Appears to be a typographical error.

Thank you, the names were corrected as your recommendation. 

iii. Line 300- Figure 4A and 4B can be merged. 

Thank you, the figure was merged in a new figure 3.

iv. Line 341- How reliable is to predict the bacterial functional profiles on the basis of 16SrRNA gene sequence data alone? Justify. Thank you. We agree with you, it seems to be very superficial. However, the inference is based on the genomic bank of the microorganism identified in the 16S rRNA. Thus, metabolic pathways of each organism are raised from the database. We included this analysis because Staph and Pseud are very different microorganisms.

v. Were only 23 (13 S. aureus and 10 P. aeruginosa) isolates obtained on culturing sputum samples? Or was that the number tested for biofilm formation? Did you isolate both S. aureus and P. aeruginosa both from a single sputum sample? Thank you, the sentence was rephrased. 

vi. The term Staphylococcus aureus and Staphylococcus spp. appears to be used quite interchangeably. Please correct the same. Same for Pseudomonas aeruginosa and Pseudomonas spp.

When we use Staphylococcus spp in the text, it is because in the analysis of 16S to cannot identify levels other than genera. However, the Group’s names were change to Staph and Pseud in the manuscript to avoid confusion.

5] Discussion is well written.

6] There are few grammatical and typographical mistakes in the manuscript which need to be carefully identified and corrected.

 

Reviewer #4:

General: Comments:It is an important non-interventional clinical trial in which sputum samples of young patients were evaluated for the relative prevalence of bacteria in their sputum. There are a lot if information collected however the way the data are presented makes the study findings and conclusions not clear and the significance of the findings not convincing. In fact, discussion of the results is stressing that the data generated simply confirms previously published studies. The analysis of the results needs a major revision after the patients are properly grouped according to their mutation status, sex and baseline FeV1. The results generated should be re-evaluated once the patients are properly divided according to their mutation status and their exacerbation status at the onset of the trial. The exacerbating patients can not be analyzed in the same group as stable patients. The authors state that FEV1 of 26% for some patients. It is unclear if this is a baseline FeV1 typical for the patients or recorded at the peak of exacerbation. If is a baseline FeV1 it would be exceptionally severe CF lung disease for so young patient (s). If this only happened for few days at the peak of exacerbation when patient experienced severe pneumonia it is completely different story. The FEV1 at onset should be measured when the patient is stable for 30 days. FeV1 can be measured also during the exacerbation of course to evaluate how severe impairment in lung function occurred during the exacerbation but the analyses of the findings should be done by separating the patients into different subgroups groups and interpretation of these findings should to be different when the stable patients are evaluated vs. when the exacerbating patients are evaluated. The exacerbating patients’ microbiota in the sputum should be evaluated separately than microbiota typical for steady state baseline microbiota typical for chronically infected patient on azithromycin and inhaled antibiotics. The way the data are presented does not brings much useful information. It is not clear what is a typical the microbiome at steady-state level in each patient and which microbiome changes occur during exacerbation.

Thanks for your constructive suggestions. We have modified the structure to clarity the text from the introduction until discussion. In addition, each question was answered in red color. Our objective was to carry out longitudinal prospective study, however the microbiota data were very different between patients. We performed various statistical analyses, without any significant result. Therefore, we opted for a different approach, try to group patients by the dominant pathogen observed in the microbiota and compare the clinical variables among these two pathogen groups, Staph and Pseud. We agree that the approach, is not common in the clinical area, however it was the only option left to take advantage of this data. Thus, this work in not a “longitudinal prospective study”. 

Is there observed during exacerbations an increase in the CFUs of specific pathogens that were already present at baseline (with the relative composition of bacteria remaining the same), or during the exacerbation the relative composition of bacteria changes dramatically?

The analyzes performed by mutation type were not carried out due to the heterogenicity of the groups, in some cases, one mutation type was observed twice. It was also not possible due to the clinical condition because statistical analyzes would not have a significant “n”.

Considering the large number of patient variables and their subclassifications, the detailed description of all these particularities results unproductive. Therefore, we chose to describe in a generic way and submit the raw data in the supplementary material.

Major specific comments:

1. Patients ages ranges 8-23 contain both pediatric cases and adult cases so should not be classified broadly as pediatric. Table containing characteristics of each patient should be included that would provide information about each patient’s age, sex, the class of the mutations that each patient had should also be stated, specific CFTR gene mutations should be stated for each patient. The table was added as supplemental material in the original submission (S1 Table), data of sex was added too. Because the information is very large to include in the main text. CRP at the time of the initiation of the trial (for stable patients with no exacerbation for 30 days), amount of exacerbation in the year prior to the study at the initiation of the study. Steady-state FeV1 recorded when there is no exacerbation. Additional data prior to the trial initiation were not collected.

2. FEV1 is only informative as a characteristic of the patient when is recorded at baseline (30 days with no exacerbation and with not major increase in CRP level compared to the previous periods without exacerbation). The patients with active exacerbation at the onset of the study should be studied separately and once they get stabilized their baseline Fev1 should be recorded. At the exacerbation the FeV1 is very volatile and can drop dramatically but it does not represent the overall FeV1 characteristics of the patient. It is affected primarily by the sudden raise in the bacterial counts and sometimes also type of combination of infections. 

This study was carried out in one of the largest cystic fibrosis centers in the state of Paraná and some patients have to travel for long distance to attend medical appointment, therefore we include as many patients as possible in the study. It would not be possible to carry out the study if we consider all the suggestions raised here. For this reason, we try to make the most of patient visits to the clinical center.

3. The biggest drop in the stable FEV1 usually occurs between age 18 and 30. It is not clear how many patients were in this age bracket? Did the authors seen a difference in baseline FeV1 and between 8–17-year-old patients compared to 18–23-year-old patients? When compared patients in this suggested cohort, the result was the following: FVC of 91 to patients among 8-17 years and FVC of 60 to patients among 18-23 years. 

4. The number of patients positive for PA, SA and both should be stated, not only the percentage of the cohort L225-L228. How many patients experienced dual infection with PA and SA? (Table S1…. check if the information is there….). We chose to describe in a generic way and submit the raw data in the supplementary material. In addition, due to the type of our study, we do not have this information. 

5. Is the relative higher abundance PA vs SA in both age group similar? According to your recommendation we have added this information in supplementary material. However, in the patients among 8-17 years the abundance was similar, on the other hand, in the patients among 18-23 years the abundance was great to Staphylococcus. 

 6. Methodology mentioned in 146-149 needs to be properly described in detail (buffers, duration of run, columns used, temperature, elution parameters, etc … ). The methodology should be written in a way that the method can be reproduced by other investigators just based on the descriptions provided. The protocol used should be included as a supplementary material. If there as space restrictions imposed by the journal, detailed protocols, including details, buffers, parameters of runs, how many times repeated, standards used should be appended as supplemental data. The way it is written is not informative at all and can not be even properly evaluated.

MALDI-TOF mass spectrometry is an equipment adapted for the diagnosis of microorganisms in the clinical routine. Samples are processed with a matrix that allows disruption and ionization of microbial proteins. Subsequently, the sample is placed on a plate with a small spot containing the isolated microorganism and placed on a plate that is then inserted by a laser into the equipment. but the equipment parameters are fixed 

7. Primers sequences should be included 5’-3’ in addition to their positions that has been mentioned in L178. Name of these primers is a universal consensus. Nomenclature is related to a specific region in the gene and sense. Thus, the form of writing is correct.

8. The information about CFTR gene mutation presented as percentages should be at least referred to the Table in the supplement that contains specific information regarding CFTR mutations each patient included in the study had, so not only percentages would be presented but the number of patients for each mutation type combination should be clear. According to your recommendation we have added this information in the supplementary material.

9. The description of BMI does not take into consideration if the patients were males or females the range of BMI changes with age and sex, some children grow faster than other children so the way the BMI is presented in not informative at all. The BMI was obtained by comparing patients of the same age. In addition, when compared patients by sex, the result was the following: BMI of 18,34 in female and BMI of 18,55 in male.

10. In the discussion, the authors state that the Staphylococcal infections are not usually treated because there is not good treatment protocol. Authors findings suggest that the Staphylococcus species tested during this study were sensitive to oxacillin and vancomycin. It is not clear if these laboratory findings were used in the study to select proper treatment for Staphylococcal exacerbation or not. This should be at least mentioned. 

Thank you for your suggestion. The sentence was rephrased to make it clearer. On the other hand, all S. aureus isolates were sensitive to oxacillin and vancomycin, lines 412. Our study did not aim to make interventions in the treatment and clinical management of patients, so the results of classical microbiology were not used to guide treatment.

11. The way the authors group the data into two groups over 50% Staphylococcus spp. or Pseudomonas spp. made the analysis of the data and resulting conclusions predictable and expected and diminished clinical significance of presented findings. Thank you very much for the observation, because of this, the limitations of our study were included in the discussion.

12. Lack of identification of specific microbe associated with apparent exacerbation is not well discussed; possibility of viral infections (frequently not diagnosed because there is no specific diagnostic test established, or/and the physician does not asks for more sophisticated tests to be done) and very slow growing or difficult to propagate in vitro bacterial infections (e.g mycobacterial species, such as M. abscessus complex). We agree that there are several other microorganisms related to the pulmonary exacerbation of patients. However, the methodology of our study does not allow specific identification of these pathogens, therefore we suggest in lines 448-452 further studies with a more specific methodology.

Minor editing errors: 

Abstract: Sentence starting from Clustering…. and the sentence starting from “Had reduced….” Should be all one sentence L52: Authors state the 48% patients were delF508 – this is not clear – 48% had one allele of delF508 or both alleles with the same delF508 mutation? Thank you, the sentence was rephrased.

---

## [Editor Report · Decision Letter 1]

9 Aug 2022

Lower airway microbiota and decreasing lung function in young Brazilian cystic fibrosis patients with pulmonary Staphylococcus and Pseudomonas infection

PONE-D-22-04509R1

Dear Dr. Mesa,

We’re pleased to inform you that your manuscript has been judged scientifically suitable for publication and will be formally accepted for publication once it meets all outstanding technical requirements.

Kind regards,

Abdelwahab Omri, Pharm B, Ph.D, Laurentian University

Academic Editor

PLOS ONE

---

## [Editor Report · Acceptance letter]

11 Aug 2022

PONE-D-22-04509R1 

Lower airway microbiota and decreasing lung function in young Brazilian cystic fibrosis patients with pulmonary *Staphylococcus* and *Pseudomonas* infection 

Dear Dr. Mesa:

I'm pleased to inform you that your manuscript has been deemed suitable for publication in PLOS ONE. Congratulations! Your manuscript is now with our production department. 

Kind regards, 

on behalf of

Dr. Abdelwahab Omri 

Academic Editor

PLOS ONE